# Enhancing Clinical Note Summarization: Iterative Reflexions with Small-model Supervision and Error2Correct Demonstrations

## Abstract

Generating clinical notes from doctor-patient dialogues is an important task in medical artificial intelligence. Mainstream methods currently employ large language models with few-shot demonstrations to tackle this challenge. However, the absence of domain knowledge supervision in these models often results in issues like missing key information, irregular writing standards, and non-compliant language styles. To this end, in this paper, we propose a novel iterative reflexion framework with small-model supervision and Error2Correct demonstrations for clinical note summarization. In this framework, we leverage a large model to produce clinical notes and design a small model trained on domain-specific data to evaluate the generated content. To enhance the quality of the generated clinical notes, we further propose Error2Correct demonstrations, which consist of error examples, error analysis, and corresponding correct examples, to help the large model identify and rectify errors effectively. To evaluate the effectiveness of our proposed method, we conduct extensive experiments on both Chinese and English datasets. The results demonstrate that our method achieves state-of-the-art performance on both datasets for the clinical note summarization task.

## 1 Introduction

Recent years have witnessed remarkable advancements in medical artificial intelligence, holding the potential to revolutionize clinical documentation records Gu et al. (2021). Extracting clinical notes from the doctor-patient dialogues is a vital part of maintaining this documentation Tang et al. (2023). While it yields valuable data for health decisions, it also places a substantial burden on clinicians Moramarco et al. (2022). Hence, it is crucial to develop efficient and robust models for automatically summarizing doctor-patient dialogues.

Currently, there are two mainstream methods for automatically generating clinical notes from doctor-patient dialogues. The first method is to fine-tune a pre-trained language model (PLM), such as BART Lewis et al. (2020) and T5 Raffel et al. (2020). These methods focus on dividing the dialogue into several segments and then generating the corresponding clinical note text Zhang et al. (2021); Krishna et al. (2021). To ensure the professionalism of the generated notes, domain-specific medical knowledge like semantic types Joshi et al. (2020); Michalopoulos et al. (2022) is also incorporated during the generation process. However, due to the limited availability of extensive annotated data, these methods may not faithfully reflect the original doctor-patient dialogue, presenting challenges when applied in practical applications. The second method leverages a large language model (LLM) with few-shot demonstrations, which is enriched with a vast repository of global knowledge. With the advent of LLMs like ChatGPT[1], researchers have harnessed the power of task-specific instructions and demonstrations to achieve better results in summarizing doctor-patient dialogues. For instance, in the recent MEDIQA-Chat clinical note generation competition[2], Giorgi et al. Giorgi et al. (2023) won the first-place position using GPT-4[3] with demonstrations.

---

[1] https://chat.openai.com/?model=chatgpt
[2] https://sites.google.com/view/mediqa2023/clinicalnlp-mediqa-chat-2023
[3] https://chat.openai.com/?model=gpt-4

This victory shows that large models (such as GPT-4) can perform well on this task, relying solely on demonstrations without fine-tuning large amounts of domain data.

However, clinical notes generated by the LLM-based methods mentioned above still face two notable problems due to the absence of domain knowledge supervision. First, they do not align with the *explicit* guidelines outlined in official medical documents. This is primarily evident in the omission of essential elements within clinical notes and inconsistencies in writing criteria. For instance, official documents typically recommend that the "Chief Complaint" section should include symptoms, their location, and their duration. However, LLM-generated results may miss critical time-related information. In addition, official guidelines require that when conveying time-related information in "Chief Complaint", it is imperative to adhere rigorously to the writing criteria for precise time. The use of ambiguous phrases, such as "1-2 days", is explicitly prohibited. Second, these clinical notes fail to meet the *implicit* industry-specific language style, particularly in Chinese clinical notes. For instance, phrases like "occasional cough" used in doctor-patient dialogue should be rephrased as "paroxysmal cough" in clinical notes.

To address the above problems, in this paper, we propose an innovative approach known as iterative **REFLE**xions with **S**mall-model supervision and **E**rror2Correct demonstrations (denoted as REFLEXSE) for clinical note summarization. Specifically, we first adopt an LLM (e.g., GPT-4) with human-crafted instruction to generate the entire clinical note as an initial dialogue summary. For each section in the initial note, we still utilize an LLM (e.g., ChatGPT) with the designed *rule prompts* and *Error2Correct* demonstrations to refine the content, ensuring it adheres to both explicit and implicit requirements. Then, we develop a small content scorer model to score the refined results produced by the LLM. Finally, we introduce an iterative scheduler to evaluate the necessity of further refinement iterations. Our method effectively addresses the above-mentioned challenges through three key characteristics. 1) Iterative reflexion. It is not enough to directly generate final clinical notes with an LLM. Hence, we introduce the idea of iterative reflexion, enabling the LLM to continuously refine its generated results. 2) Small-model supervision. Many existing reflexion frameworks rely on the LLM itself to determine the necessity of iterations. However, this strategy is not advisable in our task due to the LLM's limited medical domain knowledge. Hence, we develop a small model trained on the domain corpus to assess whether continuous iteration is necessary. 3) Error2Correct demonstrations. The reflexion process of LLMs requires clear guidance. Hence, we provide Error2Correct demonstrations, including error examples, error analysis, and corresponding correct examples, to empower large models with the ability to detect and correct errors effectively.

**Contributions.** In brief, the contributions of this paper are summarized as follows:

- We are the first to propose an iterative reflexion framework in the clinical note summarization task. The most significant characteristic of this framework is its unique collaboration between large and small models, that is, using a small model enriched with medical domain knowledge to assist LLMs in generating specialized clinical notes.

- We introduce Error2Correct demonstrations, a valuable addition that offers precise guidance for large model reflexion. This inclusion serves to continuously enhance the capacity of large models in error detection and correction.

- Experimental results on two datasets show that our model achieves state-of-the-art performance. In particular, our method outperforms the previous best method on IMCS-V2-MRG by an average of 2.97%. Additionally, it surpasses the GPT with the in-context learning (ICL) method on ACI-BENCH by 2.48% in terms of the Meteor metric.

## 2 RELATED WORK

**Clinical Note Summarization.** The techniques on this task can be divided into two categories Yim & Yetisgen-Yildiz (2021); Gao et al. (2022); Cai et al. (2022): 1) PLM-based approaches Song et al. (2020); Michalopoulos et al. (2022) and 2) LLM with few-shot demonstrations Van Veen et al. (2023). The first category is to fine-tune PLMs for clinical note summarization Krishna et al. (2020); Nair et al. (2023). For instance, the BART-based summarization model Zhang et al. (2021) is a two-stage framework by learning two fine-tuned BART models: one for summarizing dialogue chunks into partial summaries, followed by one for rewriting the collection of partial summaries into a complete summary. However, a scarcity of annotated data in specific domains poses challenges

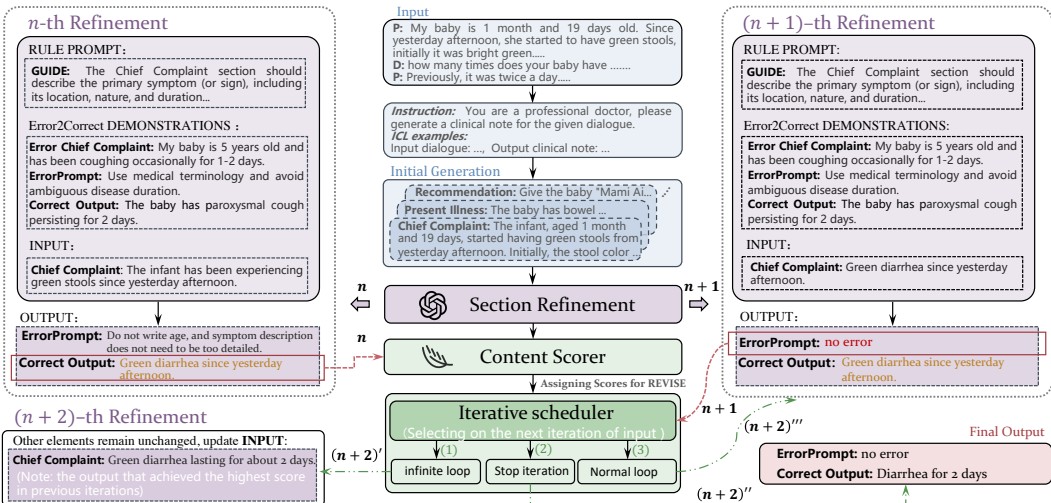

Figure 1: Our solution consists of four key steps. The first step is to create an initial complete clinical note for a given dialogue. In the second step, we adopt an LLM with the designed *rule prompt* and *Error2Correct* demonstrations to refine each section's content in the initial clinical note. The third step introduces a small content scorer model to assess and score the refined content. Finally, we design an iterative scheduler to determine the necessity for further refinement iterations.

in training high-performance models, which can lead to lower-quality generated clinical notes. The second category is to use LLMs (e.g., ChatGPT) with few-shot demonstrations Giorgi et al. (2023); Tang et al. (2023); Nair et al. (2023). The core idea of this type of method is to use the ICL method to select several demonstrations from the training set as input to LLMs, and ask them to generate the corresponding clinical note results based on these demonstrations. However, most of these methods directly generate results without domain knowledge supervision, leading to issues like missing key information, irregular writing criteria, and inconsistent language styles.

**LLM-based Reflexion.** There are two main types of mainstream LLM-based reflexion methods: 1) Knowledge-augmented methods Chern et al. (2023) and 2) self-reflexion methods Shinn et al. (2023). The first type is to inject external knowledge into the reflexion process of LLMs to reduce hallucinations and enhance result faithfulness. Typical methods along this line are LLM-AUGMENTER Peng et al. (2023) and SummIt Zhang et al. (2023). LLM-AUGMENTER is proposed to improve LLMs with task-specific knowledge and automated feedback. SummIt refines results iteratively through self-evaluation, feedback, and knowledge integration. The second type is to employ an LLM for both initial results and iterative refinements, such as Reflexion Shinn et al. (2023) and SELF-REFINE Madaan et al. (2023). Recent research explores using a separate model for evaluation and optimization Akyürek et al. (2023). However, the above methods have the following disadvantages: 1) insufficient domain knowledge in the evaluation model and 2) Lack of learning demonstrations from error examples with the analysis to correct examples.

## 3 METHODOLOGY

Given a doctor-patient dialogue, this paper aims to generate a clinical note summarizing the dialogue. The note consists of one or multiple sections, such as "Chief Complaint", "Present Disease", "Auxiliary", "Past History", "Diagnosis", and "Suggestions". As depicted in Figure 1, the solution framework for clinical note summarization mainly has the following four steps: initial generation, section refinement, content scorer, and iterative scheduler.

### 3.1 INITIAL GENERATION

Given a doctor-patient dialogue, the purpose of this process is to generate the initial clinical note. Since the quality of the initial results has a significant impact on the subsequent reflexion step's

output quality, in this paper, we adopt one of the most powerful models (i.e., GPT-4) to generate the initial results. In addition to the dialogue itself, we introduce two additional inputs to GPT-4: natural language instructions and ICL examples. For the natural language instruction, we design "You are a professional doctor, please generate the clinical note for me according to the given medical dialogue". As for ICL examples, a common practice is to pick dialogues from the training data that are similar to the input dialogue Wang et al. (2022b) and utilize them as examples for GPT-4, along with their corresponding clinical notes. However, an observed challenge with this approach is the emergence of *the copying effect* Lyu et al. (2023), where the model tends to copy snippets of these examples into the output, without paying attention to the input doctor-patient dialogue. To address this situation, we adopt a simple yet effective strategy – randomness. By randomly selecting ICL examples from the training set, we ensure that the model concentrates on generating information summaries specifically based on the doctor-patient dialogue.

## 3.2 SECTION REFINEMENT

After generating the initial clinical note, our subsequent task is to refine the content of each section within the note. In this paper, we treat each section as an independent unit and apply an identical refinement process. For convenience, we take the "Chief Complaint" section as an example to describe the specific refinement process. We utilize ChatGPT for this step instead of GPT-4, primarily because the refinement process often requires multiple iterations and leveraging GPT-4 for these iterative phases would result in prohibitively high costs. Notably, clinical note generation diverges from common text generation tasks. It necessitates consideration of both the explicit guidelines outlined in official medical documents and the implicit language style characteristic of clinical notes. Hence, in addition to the initialized content of "Chief Complaint" as input, we introduce two additional parts. The first part comprises a human-written *rule prompt*, designed to align the model with both explicit and implicit specifications. However, we observe that the model may not always strictly adhere to these given requirements. To address this, we introduce the second part, which consists of *Error2Correct* demonstrations. Each demonstration comprises three components: 1) An illustrative error example that violates "Chief Complaint" constraints (e.g., "baby's cough persists for 1-2 days"). 2) An *ErrorPrompt*, providing a descriptive explanation of the issues with the aforementioned error example (e.g., "avoid ambiguous disease duration"). 3) The correct expression of the error example (e.g., "the baby has paroxysmal cough for 2 days"). By providing such examples, the model can enhance its ability to generate high-quality "Chief Complaint" content.

## 3.3 CONTENT SCORER

The purpose of this section is to score the content refined in Section 3.2 from different aspects, such as content scope, writing criteria, and language style. Its input is the refined content of each section in the clinical note and the output is the content score. In this paper, we use an open-source medical pre-trained language model as the backbone for the content scorer. Based on this model, we leverage a linear layer to output a scalar score. For the model training, we adopt a loss function similar to the reward model in InstructGPT Ouyang et al. (2022), as shown below.

$$loss(\theta) = -E_{(y_g, y_n) \sim D}[\log(\sigma(r_\theta(y_g) - r_\theta(y_n)))], \tag{1}$$

where $r_\theta(y)$ is the scalar output of the scorer for the content $y$. For the given doctor-patient dialogue in the training set, we denote $y_g$ as the ground truth section content, and $y_n$ as the negative samples, which are the initial content generated in Section 3.1. $D$ is a set of sample pairs containing the ground truth ($y_g$) and the corresponding initialized content ($y_d$). From the above equation, we know that the model assigns higher scores to content that closely approximates the ground truth, while content closer to the initial generation receives lower scores.

Notably, 1) the initialized content may be correct in some cases, and these correct ones cannot serve as negative samples. Hence, we need to remove the correct content, keeping the remaining portions as negative samples. To determine the correct samples, we employ the strategy of length distance between the initialized content and the ground truth. If the distance is less than the predefined threshold, we consider it correct and remove it from the negative sample set. 2) We choose to exclude doctor-patient dialogues as input to the scorer model for two specific reasons. First, doctor-patient dialogues are often too lengthy, which could adversely impact the performance of the scorer model. Second, the main objective of the content scorer is to evaluate the extent to which the

content scope, writing criteria, and language style align with the guidelines specified in the official instruction. Its primary focus lies in evaluating these aspects, rather than determining the relevance of the clinical note to the corresponding dialogue, a task assigned to ChatGPT in Section 3.2.

### 3.4 ITERATIVE SCHEDULER

This part aims to empower large models to engage in iterative reflexion, ensuring the continuous optimization of the content in each section, as a single refinement is often insufficient. During the reflexion process, we design an iterative scheduler, which mainly involves three aspects: defining the input for the next iteration, developing strategies to prevent infinite loops, and establishing termination conditions for the iterations.

In the next iteration, we consider the refined section content in the clinical note from the current iteration as part of the input. Additionally, we incorporate the initially generated result in Section 3.1 and design prompts to guide the LLM to reference the initial result. This strategy is designed to prevent the large model from encountering semantic drift during the iteration process, thus reducing the risk of potential hallucinations. Notably, the input does not include the doctor-patient dialogue. This decision is driven by both length limitations and the observation that the initial results often encompass all the essential information present in the ground truth.

During the iterative process, the model may encounter the issue of getting trapped in an infinite loop, which hampers its ability to generate optimal results. In our approach, we specifically identify two situations that can lead to this occurrence. 1) The ErrorPrompt displays a result such as "no error" or "no need to modify", and the result with the highest frequency across all iterations is placed after the Top-K of all-round result scores evaluated by the scorer model. 2) The results from several consecutive iterations of the large model's reflexion remain consistent, and their scores are ranked after the Top K across all rounds of results. When the large model encounters an infinite loop, we adopt a different strategy to select the input for the next iteration. Instead of using the output of the current iteration, we choose the best output from all previous iterations as the input for the next iteration. Moreover, to avoid the model becoming trapped in the previously encountered infinite loop, we maintain a blacklist of results. When the model's output matches an entry in the blacklist, we slightly adjust the prompt, such as adding a line break or modifying the order of demonstrations, to ensure the model produces different results. In this way, we can break the infinite loop and continue the iteration process effectively.

There are three termination conditions for iterative reflexion: 1) When the ErrorPrompt generated by the LLM contains signals like "no error" or "no need to modify", and the small model's score for the result with the highest frequency is ranked in the Top-K among all iteration, we stop the reflexion process. However, if the number of categories producing different results in all iterations is less than $K$, we slightly adjust the prompt to continue with iterative reflexion. 2) If the output results of the LLM remain consistent for several consecutive iterations, and the result is ranked in the Top-K by the scorer model, we terminate the loop. 3) To prevent excessively long iterations, we set a limit on the maximum number of iterations. If the number of loops exceeds $N$, we terminate the loop.

## 4 EXPERIMENTS

In this section, we perform extensive experiments to evaluate our proposed method on public Chinese and English datasets. We also provide detailed analyses for in-depth insights into our approach.

### 4.1 EXPERIMENTAL SETUP

**Datasets.** In this paper, we employ two datasets: the Chinese dataset IMCS-V2-MRG[4] from the CBLUE benchmark Chen et al. (2023) and the English dataset ACI-BENCH[5] Yim et al. (2023). First, IMCS-V2-MRG comprises medical dialogues and the corresponding clinical notes authored by professional doctors. As of now, it stands as the sole publicly accessible Chinese dataset for this task. In this dataset, the clinical note is divided into six sections: "Chief Complaint", "Present

---

[4]https://tianchi.aliyun.com/dataset/95414

[5]https://github.com/wyim/aci-bench

Disease", "Auxiliary", "Past History", "Diagnosis", and "Suggestions". The training set contains a total of 2472 samples, while the validation set contains 833 samples. Due to the absence of ground truth in the test set, we allocate the first 200 samples from the validation set as our test set, with the remaining samples serving as the validation set. Second, ACI-BENCH offers high-quality English medical dialogues and standardized clinical note formats. It's worth noting, however, that ACI-BENCH's drawback lies in its relatively limited data volume. The dataset comprises 67 training sets, 20 validation sets, and 120 test sets. The clinical notes in this dataset are segmented into four parts: "Subjective", "Objective_exam", "Results", and "Assessment and Plan".

**Baselines.** On IMCS-V2-MRG, we compare the following baselines. First, we consider BART-Base-Chinese[6] (denoted as BART-C) and T5-Pegasus[7] as the baseline models. Additionally, we consider two-stage approaches. That is, we first employ ERNIE Sun et al. (2019) to classify each round of dialogue and remove irrelevant information, such as greetings between doctors and patients, to focus solely on medical care-related content. Then, we utilize both BART-C and T5-Pegasus for generating clinical notes. We also include IDEA-CCNL Wang et al. (2022a), which holds the impressive performance on multiple Chinese summarization tasks. Moreover, we evaluate ChatGPT and GPT-4 with ICL examples respectively that exhibit the highest similarity to the input dialogue. These two methods are denoted as ChatGPT+ICL and GPT-4+ICL, respectively.

On ACI-BENCH, we also consider BART Lewis et al. (2020) as the baseline model. On the basis of this model, we consider two variants: BioBART Yuan et al. (2022) and BART+Fine-tuning (denoted as BART+FT). BioBART is a version of BART that has been pre-trained on PubMed Cohan et al. (2018) abstracts. BART+FT is a fine-tuned version of BART on the SAMSum corpus Gliwa et al. (2019). LED-Pubmed[8] Beltagy et al. (2020) is also an encoder-decoder architecture, but it can accept longer input than BART. Besides, we compare our method with InstructGPT (Text-Davinci-002[9] and Text-Davinci-003), ChatGPT, and GPT-4. Based on GPT-4, we further incorporate ICL examples that are most similar to the input dialogue.

**Metrics.** Following to Moramarco et al. (2022), we report seven metrics: Rouge-1, Rouge-2, Rouge-L, the average of Rouge-1/2/L (denoted as Rouge-Avg), Meteor, Bertscore, and the overall average of all metrics except Rouge-Avg. A good model needs to achieve high scores across all these metrics.

**Implementations.** On IMCS-V2-MRG, we apply the proposed reflexion framework for the "Chief Complaint", "Present Disease", and "Suggestions" sections. This is because the contents of the remaining sections can be either easily summarized or left empty. Nevertheless, during the final metric computation, we still calculate metrics for all sections, with other sections utilizing the initialized summary as the ultimate results. We set the number of demonstrations to 2 and use a temperature parameter of 0.2. All other hyperparameters of the OpenAI API are maintained at their default values. To ensure reliable results, we conduct three replicate experiments and calculate the average value. For the content scorer model, we employ the Chinese medical pre-training language model, ERNIE-HEALTH-Chinese Wang et al. (2021). We fine-tune this model using 80 training samples and 10 validation samples (for selecting the best parameters). On ACI-BENCH, we compute the metrics with the reflexion results of the subsection "History of Present illness" in "Subjective" and the section "Assessment and Plan", and the initial results of the remaining sections. For the scorer model, we use LED-PubMed, an English medical pre-training language model, and fine-tune it with 40 training samples and 10 validation samples. We set $K = 2$ for both Chinese and English datasets.

## 4.2 MAIN RESULTS

In this section, we conduct a comparative analysis of our method against all baseline approaches on IMCS-V2-MRG and ACI-BENCH. The results are presented in Tables 1 and 2.

From the tables, we have the following conclusion. 1) Our method achieves state-of-the-art (SoTA) performance across most metrics, with the exception of Rouge-L, where it lags behind IDEA-CCNL by 1.84% on IMCS-V2-MRG. This difference can be attributed to the unique characteristics of Chinese clinical notes, known for their brevity and conciseness. Models trained through extensive pre-

---

[6]https://huggingface.co/fnlp/bart-base-chinese

[7]https://github.com/SunnyGJing/t5-pegasus-chinese

[8]https://huggingface.co/patrickvonplaten/led-large-16384-pubmed

[9]https://platform.openai.com/docs/models

Table 1: Model comparison (%) on IMCS-V2-MRG. The values in brackets indicate the standard deviation of the results.

| Method | Rouge-1 | Rouge-2 | Rouge-L | Rouge-Avg | Meteor | Bertscore | Overall Avg |
|---|---|---|---|---|---|---|---|
| BART-C | 51.13 | 32.58 | 47.78 | 43.83 | 22.49 | 75.48 | 45.89 |
| T5-Pegasus | 52.69 | 33.80 | 49.36 | 45.28 | 23.72 | 76.44 | 47.20 |
| EINIE+BART-C | 51.26 | 32.87 | 48.56 | 44.23 | 22.26 | 75.64 | 46.12 |
| ERNIE+T5-Pegasus | 53.95 | 35.12 | 50.67 | 46.58 | 24.44 | 76.80 | 48.20 |
| IDEA-CCNL | 55.18 | 39.71 | **51.60** | 48.83 | 26.44 | 78.49 | 50.28 |
| ChatGPT+ICL | 51.18 | 32.53 | 43.05 | 42.25 | 35.07 | 75.64 | 47.49 |
| GPT-4+ICL | 52.13 | 33.72 | 44.70 | 43.52 | 37.75 | 75.71 | 48.80 |
| REFLEXSE | **58.42** (0.10) | **39.86** (0.11) | 49.76 (0.18) | **49.35** (0.11) | **39.44** (0.13) | **78.75** (0.08) | **53.25** (0.12) |

Table 2: Model comparison (%) on ACI-BENCH.

| Method | Rouge-1 | Rouge-2 | Rouge-L | Rouge-Avg | Meteor | Bertscore | Overall Avg |
|---|---|---|---|---|---|---|---|
| BART | 49.19 | 20.84 | 24.32 | 31.45 | 35.45 | 63.02 | 38.56 |
| BioBART | 45.81 | 18.40 | 23.34 | 29.18 | 31.09 | 62.73 | 36.27 |
| BART+FT | 47.25 | 19.08 | 22.70 | 29.68 | 33.85 | 61.63 | 36.90 |
| LED-Pubmed | 30.91 | 7.27 | 11.22 | 16.47 | 29.28 | 55.19 | 26.77 |
| Text-Davinci-002 | 40.49 | 17.71 | 24.82 | 27.67 | 23.11 | 63.47 | 33.92 |
| Text-Davinci-003 | 44.43 | 21.47 | 29.48 | 31.80 | 26.43 | 66.24 | 37.61 |
| ChatGPT | 45.27 | 18.31 | 25.76 | 29.78 | 27.42 | 65.02 | 36.36 |
| GPT-4 | 51.10 | 21.97 | 29.55 | 34.21 | 33.83 | 67.69 | 40.83 |
| GPT-4+ICL | 59.01 | 28.92 | **37.27** | 41.73 | 43.97 | 72.02 | 48.24 |
| REFLEXSE | **59.36** (0.09) | **29.28** (0.13) | 36.89 (0.10) | **41.84** (0.17) | **46.45** (0.11) | **72.12** (0.12) | **48.82** (0.12) |

training and fine-tuning on large datasets are good at capturing this conciseness, thereby benefiting the computation of the Rouge-L metric. In contrast, larger models relying on few-shot ICL tend to generate more extensive content, which affects their Rouge-L scores. However, it's important to note that our method achieves a significantly higher overall average score compared to all baseline models. For example, REFLEXSE outperforms IDEA-CCNL by 2.97% on IMCS-V2-MRG and surpasses GPT-4+ICL by 0.54% on ACI-BENCH. 2) Compared to LLM+ICL, our proposed reflexion framework has shown significant improvements, particularly on Rouge-1/2 of IMCS-V2-MRG and Meteor of ACI-BENCH. As previously discussed, Chinese clinical notes are characterized by their condensed language bias, while English clinical notes feature richer information bias. The former facilitates the calculation of Rouge-1/2 metrics, primarily centered around precision, while the latter is conducive to Meteor metric calculations, primarily emphasizing recall. 3) When compared to models relying on pre-training and fine-tuning techniques, LLMs demonstrate distinct advantages in the Meteor metric over the Rouge-1/2 metrics. This distinction arises from the fact that large models tend to produce more comprehensive text information, but they may also introduce colloquial vocabulary, leading to a more significant increase in recall than precision. However, our method achieves SoTA performance across these metrics simultaneously. This achievement underscores the efficacy of our reflexion framework in standardizing medical record terminology while preserving vital information, thus ensuring a balance between precision and recall.

## 4.3 DETAILED ANALYSIS

In this section, we analyze the important components of our method in detail. Additionally, we also provide in-depth case studies.

**Accuracy of the small model.** In this paper, we introduce a small model to score the results generated at each iteration. To evaluate the accuracy of the small model, we conduct experiments on IMCS-V2-MRG and ACI-BENCH. In these experiments, if the score of the ground truth for a particular section surpasses the score of its initial generated result, we classify it as correct; otherwise, we label it as wrong. The experimental results are shown in Figure 2. These results reveal that the small model consistently achieves high accuracy across all sections in both datasets,

Table 3: Analysis small-model supervision and Error2Correct demonstrations on IMCS-V2-MRG.

| Method | Rouge-1 | Rouge-2 | Rouge-L | Rouge-Avg | Meteor | Bertscore | Overall Avg |
|---|---|---|---|---|---|---|---|
| | | | *Small-model Supervision* | | | | |
| Variant 1 | 57.49 (0.01) | 38.28 (0.12) | 48.67 (0.01) | 48.15 (0.04) | 39.33 (0.01) | 77.99 (0.05) | 52.35 (0.04) |
| Variant 2 | 57.57 (0.08) | 38.51 (0.10) | 48.78 (0.20) | 48.27 (0.12) | 39.43 (0.03) | 78.10 (0.13) | 52.48 (0.11) |
| Variant 3 | 57.66 (0.09) | 38.69 (0.11) | 49.16 (0.08) | 48.50 (0.12) | 39.50 (0.10) | 78.23 (0.03) | 52.65 (0.09) |
| | | | *Error2Correct Demonstrations* | | | | |
| Variant 4 | 56.24 (0.16) | 36.81 (0.04) | 47.67 (0.02) | 46.91 (0.10) | 38.17 (0.09) | 77.50 (0.07) | 51.28 (0.08) |
| Variant 5 | 56.29 (0.06) | 37.23 (0.12) | 47.37 (0.04) | 46.96 (0.03) | 38.50 (0.11) | 77.38 (0.10) | 51.35 (0.09) |
| REFLEXSE | **58.42** (0.10) | **39.86** (0.11) | **49.76** (0.18) | **49.35** (0.11) | **39.44** (0.13) | **78.75** (0.08) | **53.25** (0.12) |

indicating the effectiveness of our specially designed small model.

**Effectiveness of small-model supervision.** The small model is mainly used in the "content scorer" and "iteration scheduler" steps. To validate the effectiveness of small-model supervision, we design three experiments. Experiment 1 (referred to as Variant 1): We use LLM with Error2Correct demonstrations but exclude both iterations and the small content scorer model. Experiment 2 (Variant 2): Building upon the foundation of Experiment 1, we fix the

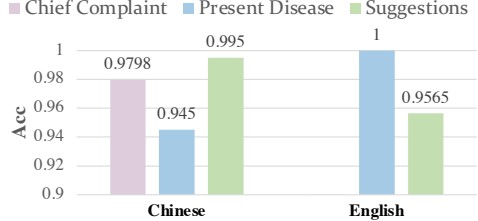

Figure 2: Accuracy of the small model.

number of iterations to 5. Experiment 3 (Variant 3): Similarly, based on Experiment 1, we adopt an iterative reflexion strategy until three distinct results occur. Then, we employ the small model to select the final output with the highest score from the three rounds of results. The experimental results are reported in Table 3.

From the results, we observe that when we limit the iteration count to 1 and exclude the small model, Variant 1 shows a significant decline in all metrics compared to REFLEXSE. Furthermore, even when we increase the iteration count to 5 in Variant 2, we do not notice a substantial improvement compared to Variant 1. This highlights that during the iterative reflexion process, significant improvements are not solely attributed to increasing the number of iterations; the presence of the small content scorer model plays a crucial role. Additionally, the strategy of employing the small model to select answers post-generation is not as effective as having the small model actively participate in the iterative process. There are two main reasons for this phenomenon. First, stepping out of the iterative infinite loop allows for the generation of more diverse and effective results. Second, some of the best results emerge after multiple iterations, typically beyond the fourth round or more. Imposing a requirement for a larger number of reflexion results as termination conditions for these smaller samples would significantly increase costs. In contrast, integrating small models into the iterative process provides an efficient solution to this issue.

**Effectiveness of Error2Correct demonstrations.** In this paper, we propose Error2Correct demonstrations to guide ChatGPT to generate high-quality clinical notes. In order to verify the effectiveness of this strategy, we design two experiments. Experiment 4 (Variant 4): We use 2 Error2Correct demonstrations but omitted ErrorPrompt. Experiment 5 (Variant 5): We use an Error2Correct demonstration. The experimental results are presented in Table 3. From the results, we observe a significant decrease in the quality of clinical notes generated by LLMs when ErrorPrompts are omitted, highlighting the importance of providing large models with insights into error causation. Additionally, a smaller number of demonstrations also have a negative impact on the model's performance. This suggests that a larger number of demonstrations can offer a wider range of reference examples for large models. However, due to model input length limitations and cost considerations, we restrict the number of demonstrations to 2 in the experiment.

**Analysis of convergence iterations for the reflexion framework.** Figure 3 illustrates the convergence iterations (C-iterations) of all test samples using our proposed reflexion solution. From the results, we observe that the results generated in the "Chief Complaint" section tend to converge within the first or second C-iteration, while those in the "Present Disease" and "Suggestions" sec-

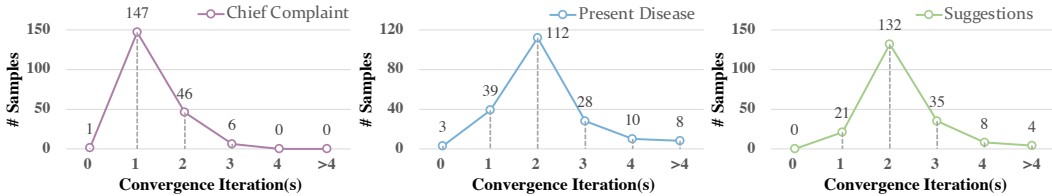

Figure 3: Analysis of convergence iterations for the reflexion framework. C-iterations=$i$ indicates the generation of $i$ distinct results in the current iteration and all previous iterations. When C-iteration=0, the result corresponds to the initial one.

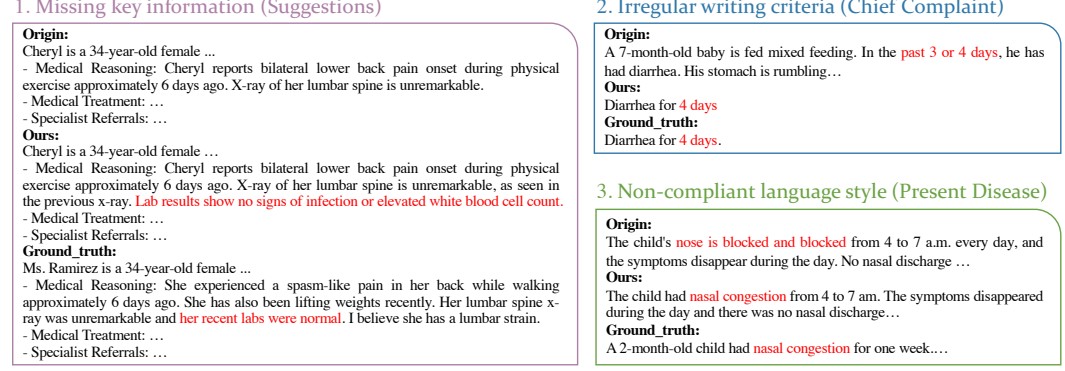

Figure 4: Case study. The words in red indicate the parts that need to be compared.

tions often reach convergence in the first, second, or third C-iteration, occasionally even extending to the fourth or beyond. This phenomenon emerges due to the relative brevity of input and output in the "Chief Complaint" section compared to the more extensive content in the other two sections. As a result, the reflexion process in the former is inherently simpler. Consequently, establishing a fixed number of rounds and outcomes during the reflexion process is not a practical approach. Instead, employing a small model alongside the LLM enables an adaptive approach, ensuring that the best results are achieved within an appropriate number of iterations.

**Case study.** To provide a more intuitive demonstration of the effectiveness of our reflexion framework, we present case studies that include the initially generated results, the results after reflexion, and the ground truth. These cases are illustrated in Figure 4. In the "Suggestion" section, we observe that key information is missing from the initially generated results. However, after reflexion, we observe the incorporation of this missing information. In the "Chief Complaint" section, our optimization efforts successfully remove a significant amount of redundant information from the clinical note, resulting in a more concise and specific representation. Additionally, we transform vague references to time, such as "past 3 or 4 days", into precise numerical values. In the "Present Disease" section, we notice a transition from an informal language, such as "nose is blocked and blocked", to a more formal and medically accurate term, such as "nasal congestion", as a result of our reflexion framework.

## 5    CONCLUSION

This paper proposes a novel iterative reflexion framework with small-model supervision and Error2Correct demonstrations for clinical note summarization. The core idea of this framework is to assign the task of generating clinical notes to a large model, while a small model, trained on domain-specific data, is tasked with evaluating the generated content. To enhance the performance of LLMs in producing high-quality results, we design Error2Correct demonstrations, including error examples, error analysis, and corresponding correct examples, to empower LLMs with the ability to detect and correct errors effectively. The experiments conducted on both Chinese and English datasets demonstrate that our proposed method achieves SoTA results. Furthermore, a detailed analysis confirms the effectiveness of the key components of our approach.

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
