# OpenReview forum: "Enhancing Clinical Note Summarization: Iterative Reflexions with Small-model Supervision and Error2Correct Demonstrations"
_ICLR.cc/2024/Conference — ICLR 2024 Conference Withdrawn Submission_

### Official Review · Reviewer_vmuM · 2023-10-27

**Soundness:** 2 fair
**Presentation:** 2 fair
**Contribution:** 1 poor
**Rating:** 3
**Confidence:** 3

**Summary:**

This paper introduces a method to enhance clinical note summarization using LLMs. The authors propose an iterative reflexion framework, which uses LLMs for initial note creation and iteratively revises it with a small scorer model and some "Error2Correct" demonstrations. Experimental results on both English and Chinese datasets show their method achieves state-of-the-art performance.

**Strengths:**

1. The authors proposed a novel iterative reflexion framework for generating the summary of medical conversations with LLMs. In the framework, they also introduced a smaller content scorer for reducing the inference cost, and curated some Error2Correct demonstrations for revising the generated content.
2. The authors have conducted experiments on both Chinese and English datasets, and experimental results show performance improvement by the proposed method.

**Weaknesses:**

1. **No human evaluation.** One of the main issues for this paper is that the evaluation of the generated summaries only involves automatic scores. However, it is well known that these scores do not often correlate with human ratings. As such, it is vital for this kind of study to have human evaluations on the quality of the generated summaries, perhaps among different aspects.
2. **Marginal improvement but significantly more cost.** From Tables 2 and 3, it seems that the proposed method only marginally outperforms the previous SOTA, especially on the English dataset (48.82 v.s. 48.24 by GPT-4 + ICL). However, the proposed method obviously uses much more computation. I would like to see more analyses on comparing the latency or computing cost by different methods.
3. Overall this is an over-optimized method for a specialized task (generating doctor-patient conversation summaries), so it might not appeal to most of the audience in this venue.

**Questions:**

1. Figure 1 is hard to interpret: normally there is a sequential order, but the steps in Figure 1 seem quite entangled, and I cannot figure out the correct step order. Could you revise it?
2. I cannot find any code or resources. Will these be publicly available?

---

### Official Review · Reviewer_Xmzo · 2023-10-31

**Soundness:** 3 good
**Presentation:** 4 excellent
**Contribution:** 3 good
**Rating:** 8
**Confidence:** 5

**Summary:**

This paper proposes an iterative reflexion framework for improving summaries generated by large language models  on the task of clinical dialogue summarization. Different from existing approaches that involve self-evaluation where LLM is used to critique and improve its own generation results, a separate small-scale fine-tuned scoring model and a separate refinement LLM are employed to iteratively score and refine a clinical summary initially generated by LLM with in-context learning. The framework applied to initial summaries generated by GPT4 + ICL shows solid improvement over several competitive fine-tuned and zero-shot ICL baselines. And the ablation studies conducted by the authors showed the necessity of both added models and several other key designs in the proposed framework. This paper overall presents an easy-to-follow yet effective refinement approach to improve upon existing LLM models in the domain of medical summarization.

**Strengths:**

Originality:
Although individual components in the proposed REFLEXSE framework has been studied under different contexts, the unique combination of the components, the design of the iterative use of the components, and the application in the medical domain are definitely original.

Quality:
The paper presents solid research with clear outline of problem, data, method, and comprehensive analysis that presents evidences for the effectiveness of proposed iterative refinement process and the necessity of component modules in the design.

Clarity:
The writing is very easy to follow and the paper is well organized. A few minor clarifications may be needed (addressed in weaknesses and questions sections).

Significance:
The paper is definitely of great interest for practical research in the domain of medical NLP, specifically on the topic of medical dialogue summarization, which has already seen a slew of applications in commercial use (e.g., AWS Healthscribe, Microsoft/Nuance DAX); but the general framework of iteratively improving upon LLM capabilities for summarization could be generalizable to text summarization in other domains as well, especially when there are specific rules for target text.

**Weaknesses:**

The major weakness is the lack of human evaluation, or more thorough selection of automatic evaluation metrics that correlate well directly with the quality dimensions proposed by the authors: the authors preamble the motivation for the research as LLM fails to follow explicit and implicit guidelines specific to the problem of clinical note generation, with which the reviewer agrees, but unfortunately the final improvement was reported only in a limited selection of evaluation metrics, most of which are based on lexical similarity (with the exception of Bertscore). It is therefore hard to draw the conclusion whether an improvement in Rouge or Meteor score is indeed an improvement in the generated summaries better meeting the said guidelines.

Another obvious weakness is the inclusion of all sections in final evaluation while applying the proposed framework only to a subset of sections (e.g., on ACI-Bench, evaluation is done for all sections but REFLEXSE is applied only to HPI and A/P). Although ablation study presented in the paper hints at the effectiveness of the framework even if only affected sections were evaluated, it would still be more robust to separate the sections not involved in the framework in the evaluation.

**Questions:**

A couple of suggestions:
1. Section 4.2, *...large datasets are good at capturing this conciseness, thereby benefiting the computation of the Rouge-L metric...* may be an unsubstantiated statement, the different between the leading Rouge-L score and the model's score is barely larger than one s.d., the difference may just be statistically insignificant.
2. Section 4.2, *...The former facilitates the calculation of Rouge-1/2 metrics, primarily centered around precision, while the latter is conducive to Meteor metric calculations, primarily emphasizing recall...*. Can't say I agree with the statement here: what type of Rouge scores did you report in the results? Rouge P/R/F1? Unless it is Rouge precision, rouge is generally a recall-based metrics. Meteor in its default setting is biasing towards recall, but a low meteor score could be caused by either low recall of unigram or larger penalty of chunk continuity (e.g., addition of unmatched words or wrong word order). Are any of these factors more prominent in the generated summaries than precision/recall based analysis?
   1. it may be instructive to report the typical length (word count) of the generated summary, or an average ratio in word count between generated and reference summaries. Rouge is well-known to exhibit a length bias, so the different improvement across different metrics may simply be due to length.
3. Section 4.3, **Accuracy of the small model**, since you also mentioned some initial generation may be correct and removed from fine-tuning the small scorer model, using a score comparison to define accuracy is a bit contradictory. One interesting quantity to analyze may be $Corr(X, Y)$, where X is the difference in the scores between reference summary and a generated summary (not necessarily the initial generation), and Y is some metrics of the generated summary evaluated against reference. If you observe a strong negative correlation, then it means the scoring model $r()$ correlates well with the target metrics, therefore making it appropriate as a "judge" for iterative scheduler that aims to improve the target metrics.
4. Figure 2, it is better to use separate legends for the two bars for the ACI-Bench dataset, since the section names are different across these two datasets.

A few clarifying questions:
1. How are the examples in Error2Correct module selected? Randomly chosen or is some similarity-based selection involved? How are they created? Are they available in the datasets (probably not in ACI-Bench) or annotated through separate effort?
2. How does the Rule prompt embed the guidelines for clinical notes? Take SOAP note for example, specific institutions may provide very specific guidelines such as "always use pronouns", "always state date in MM/DD/YYYY format", etc. Did you use any instructions like these, or a more generic collection of guidelines. Can you provide a table of rule prompts used in the experiments?
3. Table 3, variants 1-3 very clearly show the importance of the small-model, but variant 1 is one iteration of refinement through error correction and it's already showing large improvement over GPT+ICL baseline, this makes me wonder if you could provide a plot of model performance as a function of inference time or number of iterations to show if the framework already displays diminishing return behavior. In other words, how much extra time would one pay to get improvement from (Rouge1 52.13 -> 57.49 to Rouge1 52.13 -> 58.42)?

Lastly, a very minor suggestion is that since clinical data is involved, it's best to include an "Ethical Consideration" section to state clearly the nature of the data and no PHI was involved.

---

### Official Review · Reviewer_T7JA · 2023-10-31

**Soundness:** 2 fair
**Presentation:** 1 poor
**Contribution:** 2 fair
**Rating:** 3
**Confidence:** 4

**Summary:**

The paper contributes to methods to summarise clinical language in Chinese and English. It introduces a new method and compares it experimentally with two baselines on two datasets.

**Strengths:**

The considered methods (e.g., Chat Generative Pre-trained Transformer and Bidirectional and Auto-Regressive Transformers) are trending topics.

Applications to health and medicine are societally of utmost importance.

The paper is quite carefully written.

**Weaknesses:**

Although scoping the review of related work to address only the most recent methods is justifiable, the paper is lacking in its acknowledgement and appreciation of prior text summarisation work. For example, please cite some systematic reviews of clinical and biomedical text summarisation studies. E.g., the following papers may be helpful:

Mishra, R., Bian, J., Fiszman, M., Weir, C. R., Jonnalagadda, S., Mostafa, J., & Del Fiol, G. (2014). Text summarization in the biomedical domain: a systematic review of recent research. Journal of biomedical informatics, 52, 457–467. https://doi.org/10.1016/j.jbi.2014.06.009

Pivovarov, R., & Elhadad, N. (2015). Automated methods for the summarization of electronic health records. Journal of the American Medical Informatics Association : JAMIA, 22(5), 938–947. https://doi.org/10.1093/jamia/ocv032

Wang, M., Wang, M., Yu, F., Yang, Y., Walker, J., & Mostafa, J. (2021). A systematic review of automatic text summarization for biomedical literature and EHRs. Journal of the American Medical Informatics Association : JAMIA, 28(10), 2287–2297. https://doi.org/10.1093/jamia/ocab143

The paper should clarify already earlier that it addresses text summarisation in Chinese and English.

Addressing research ethics is essential in health and medicine. I could not find a statement about obtaining the proper approvals and permissions of using the electronic health data for the purposes of this paper.

Although performance evaluation results are included, I could not find information about statistical analyses (e.g., significance tests or confidence intervals) that would have been conducted as part of this study. Please clarify if such analysis results are to be derived by the reader from the means and standard deviations (e.g., Tables 1-3).

Some more attention to detail could be paid. For example, make sure that all acronyms are introduced, check the in-text citation style (i.e., citep vs. citet if using LaTeX), make sure all images are easy to read (e.g., text should not be much smaller than the font used in figure captions), and spend some time perfecting the list of references.

**Questions:**

How was research ethics addressed in this study? What possible approvals and permissions were obtained for accessing and using the datasets for the purposes of this paper? What benefits and risks were identified in conducting this research? What are the envisioned positive or negative impacts of this study going forward?

How does this study relate to and contribute to the broader body of text simplification work? Are the results limited to Chinese and English? In what ways does the study facilitate, e.g., the clinical language processing or applied AI communities?

Were the findings statistically and/or practically significant? How was this assessed?

**Details Of Ethics Concerns:**

Addressing research ethics is essential in health and medicine. I could not find a statement about obtaining the proper approvals and permissions of using the electronic health data for the purposes of this paper.

---

### Official Review · Reviewer_Sjio · 2023-11-01

**Soundness:** 4 excellent
**Presentation:** 1 poor
**Contribution:** 3 good
**Rating:** 3
**Confidence:** 5

**Summary:**

The work focuses on iteratively improving the summaries (EHR notes) generated from the LLM. They do it by using several steps, Error demonstrations, Error Prompt, a separate content scorer, avoiding semantic drift, and a strategy to pick the best summary / avoiding too many iterations.

**Strengths:**

The work describes an elaborate strategy to improve clinical note summarization using LLM and a small model.
The way the small local model is incorporated to score the LLM's output in a loop is interesting and is a relevant area of research.
The work has evaluated different modules in their system separately and has justified their inclusion with quantitative results.

**Weaknesses:**

The presentation should be improved:
- A lof description about the system working in iterations isn't clear.
- A brief description of the system proposed before diving into the specifics of each of the module would help get better clarity. Figure 1 provides a good overview. The authors should describe the figure in detail in the paper and better use it in the description of their system.
- ITERATIVE SCHEDULER subsection isn't clear.
- The role of the content scorer and how the authors get multiple outputs is also unclear.
- More details on how the content scorer is trained are required, for example, it isn't clear how the negative samples are generated from the dataset. Does the dataset has paired data or were the negative sample generated by the LLM itself?

Although the variations and baselines were rigorous and numerous, a variation of their work with an LLM scorer (with few-shot) should have been compared against the proposed small model scorer. Especially because the authors claim LLM might not have the necessary medical knowledge to do the task.

The dataset should have been described with some stats like the average summary/dialog/note lengths to ground the readers.


My main concern is the readability of the paper. I would consider a higher score if the readability is improved.

**Questions:**

The paper references the same paper multiple times "Generating soap notes from doctor-patient conversations using modular summarization techniques."

---

### Comment · Reviewer_T7JA · 2023-11-20
**Summary**

Based on the four reviews and having not received author response, I cannot support accepting the paper. I do not see a reason to revise my original review either.

---

### Meta-Review · Area_Chair_mYeC · 2023-12-06

**Metareview:**

The authors consider the problem of generating clinical notes from patient-provider conversations. They observe that using LLMs "off the shelf" may fail to comply with guidelines (related to both formatting and content). To address this, they propose using a second, smaller, supervised model to correct outputs generated by a (very) large model. While reviewers agreed that the proposed framework itself constitutes a potentially interesting direction for exploration, this particular work is weakened by a lack of manual evaluations, presentation issues, and appropriate contextualization in light of related work (much of which the authors appear to have overlooked).

**Justification For Why Not Higher Score:**

The main issues here are: The evaluation, presentation issues which make it difficult to follow some key aspects of the work, and treatment of relevant prior work.

**Justification For Why Not Lower Score:**

N/A

---

### Decision · Program_Chairs · 2024-01-16

Reject